# Epidemiology and aetiology of moderate to severe diarrhoea in hospitalised patients ≥5 years old living with HIV in South Africa, 2018–2021: A case-control analysis

Siobhan L. Johnstone[1,2]*, Linda Erasmus[1], Juno Thomas[1], Michelle J. Groome[3,4], Nicolette M. du Plessis[5], Theunis Avenant[5], Maryke de Villiers[6], Nicola A. Page[1,7]

1 Center for Enteric Diseases, National Institute for Communicable Diseases, A Division of the National Health Laboratory Service (NHLS), Johannesburg, South Africa, 2 School of Public Health, Faculty of Health Sciences, University of the Witwatersrand, Johannesburg, South Africa, 3 Division of Public Health Surveillance and Response, National Institute for Communicable Diseases, A Division of the National Health Laboratory Service (NHLS), Johannesburg, South Africa, 4 School of Pathology, Faculty of Health Sciences, University of the Witwatersrand, Johannesburg, South Africa, 5 Department of Paediatrics, Kalafong Provincial Tertiary Hospital, Faculty of Health Sciences, University of Pretoria, Pretoria, South Africa, 6 Department of Internal Medicine, Kalafong Provincial Tertiary Hospital, Faculty of Health Sciences, University of Pretoria, Pretoria, South Africa, 7 Department of Medical Virology, Faculty of Health Sciences, University of Pretoria, Pretoria, Arcadia, South Africa

* siobhanj@nicd.ac.za, siobhanjsa@gmail.com

**Data Availability Statement:** The dataset has been made available in the supplementary material.

## Abstract

Diarrhoea is a recognised complication of HIV-infection, yet there are limited local aetiological data in this high-risk group. These data are important for informing public health interventions and updating diagnostic and treatment guidelines. This study aimed to determine the pathogenic causes of diarrhoeal admissions in people living with HIV (PLHIV) compared to hospital controls between July 2018 and November 2021. Admitted diarrhoeal cases (n = 243) and non-diarrhoeal hospital controls (n = 101) ≥5 years of age were enrolled at Kalafong, Mapulaneng and Matikwana hospitals. Stool specimens/rectal swabs were collected and pathogen screening was performed on multiple platforms. Differences in pathogen detections between cases and controls, stratified by HIV status, were investigated. The majority (n = 164, 67.5%) of enrolled diarrhoeal cases with known HIV status were HIV-infected. Pathogens could be detected in 66.3% (n = 228) of specimens, with significantly higher detection in cases compared to controls (72.8% versus 50.5%, *p*<0.001). Amongst PLHIV, prevalence of *Cystoisospora* spp. was significantly higher in cases than controls (17.7% versus 0.0%, *p* = 0.028), while *Schistosoma* was detected more often in controls than cases (17.4% versus 2.4%, *p* = 0.009). Amongst the HIV-uninfected participants, prevalence of *Shigella* spp., *Salmonella* spp. and *Helicobacter pylori* was significantly higher in cases compared to controls (36.7% versus 12.0%, *p* = 0.002; 11.4% versus 0.0%, *p* = 0.012; 10.1% versus 0.0%, *p* = 0.023). Diarrhoeal aetiology differed by HIV status, with *Shigella* spp. (36.7%) and *Salmonella* spp. (11.4%) having the highest prevalence amongst HIV-uninfected cases and *Shigella* spp. (18.3%), *Cystoisospora* (17.7%), and

---

**Funding:** The ANDEMIA study was supported by the German Federal Ministry of Education and Research (https://www.bmbf.de/bmbf/en/home/home_node.html) [Grant number 81203616] to N. A.P and S.L.J. The funders had no role in study design, data collection and analysis, decision to publish, or preparation of the manuscript.

**Competing interests:** The authors have declared that no competing interests exist.

*Cryptosporidium* spp. (15.9%) having the highest prevalence in cases amongst PLHIV. These differences should be considered for the development of diagnostic and treatment guidelines.

## Introduction

Diarrhoeal diseases pose a significant health burden across the globe, causing over 1.6 million deaths annually amongst all ages [1]. The majority of these deaths occur in low- and middle-income countries (LMICs), specifically in south Asia and Sub-Saharan Africa (SSA) [1]. Despite a decrease in global diarrhoeal mortality rates over the past two decades, decreases have not been uniform across age groups or settings [2]. Unsafe water, poor sanitation and hygiene (WaSH) [1], as well as malnutrition and compromised immunity [3] are the main risk factors associated with diarrhoeal morbidity and mortality. Other risk factors identified in LMICs include residence in a slum, use of communal toilets and exposure to animals [4]. Children under the age of 5 years are at the highest risk, however there is a significant burden in older age groups, specifically in LMICs [1, 5]. With global improvements in access to safe drinking water [6], much of the remaining burden of diarrhoeal disease may be attributed to foodborne pathogens [7]. Food safety is of specific concern in LMICs where there is increased consumption of unsafe foods, use of effluent in agriculture and changes in food distribution networks with bulk production and increased distance between production and consumption as well as large, often poorly regulated, informal sectors [7].

Studies from high-income countries (HICs) have shown the majority of diarrhoeal diseases in adults seeking healthcare to be due to *Campylobacter* spp., *Salmonella* spp., norovirus and rotavirus [8, 9]. Aetiology differs in SSA, with a meta-analysis amongst all ages identifying the main causative pathogens as *Escherichia coli*, *Cryptosporidium*, *Cyclospora*, *Entamoeba* and *Shigella* spp. [10] There are several reasons for these differences in aetiology including different exposures and risk factors associated with poorer living conditions and poor underlying health. One important consideration is the high prevalence of human immunodeficiency virus (HIV) in SSA [11]. HIV is a known risk factor for diarrhoea, with episodes in PLHIV more likely to be severe, prolonged and result in hospitalisation with higher mortality rates than in HIV-uninfected individuals [12, 13]. The aetiology of diarrhoea in PLHIV is known to shift with HIV disease progression and treatment [14]. During early disease stages, while CD4 + counts are high and viral loads low, diarrhoea is often related to HIV seroconversion. As disease progresses, diarrhoea is frequently caused by opportunistic pathogens related to decreased immune function [14]. These pathogens include bacterial infections, such as *Mycobacteria*, parasitic infections, such as *Cystoisospora belli*, *Cyclospora*, *Strongyloides*, *Cryptosporidia*, *Microsporidia* and viral infections, such as cytomegalovirus (CMV) [14–16]. Patients with low CD4+ cell counts are at increased risk for chronic diarrhoea [4] and polyparasitic infection [17]. The HIV infection itself may be responsible for a proportion of the pathogen-negative cases, through HIV enteropathy and permanent damage to the gastrointestinal mucosa [18], as well as changes in gut microbial populations leading to dysbiosis [19]. Mucosal damage reduces small bowel villous surface area, causing diarrhoea through malabsorption [20] as well as defects to cellular and humoral defence mechanisms in the gastrointestinal tract [21]. Pathogen-negative cases associated with advanced HIV disease may also be related to gastrointestinal tract involvement by *Mycobacterium tuberculosis* and/or *avium* [22] which is difficult to diagnose. A Dutch study found that 34% of diarrhoeal cases in patients with untreated HIV could not be explained by a known causative agent, and hypothesized that these unexplained

cases were due to the HIV-infection itself [23]. Initiation of antiretroviral treatment (ART), and accompanying increase in CD4+ cell count, is associated with a shift in the aetiology of diarrhoea from opportunistic infectious causes to non-infectious causes [24]. Data from both HICs and SSA show that although patients initiated on ART have significantly reduced diarrhoeal incidence compared with those not on treatment, the overall diarrhoeal incidence remains high [18, 25]. A proportion of the remaining episodes in patients with well-controlled HIV may be drug-related [18].

There are limited aetiological data on HIV-related diarrhoeal diseases in SSA, with the majority of published studies focusing on children under the age of five years, and available data in older children and adults being limited by diagnostic technology [14]. A review of laboratory records in Botswana indicated that only 14% of stool specimens submitted for routine diagnostic testing for all ages had a pathogen detected, of which 8% were bacteria and 6% parasites [26]. *Shigella* spp. and *Salmonella* spp. were the most commonly detected bacteria while *Cystoisospora* spp. and *Cryptosporidium* spp. were the most commonly detected parasites. Viral testing was not done. Although HIV status was not available for the participants in this analysis, Botswana is a high HIV prevalence setting with an estimated 23.9% of the population between 15–49 years old living with HIV infection during the study time period [26]. Contrary to these findings, a longitudinal cohort study in Zambian adults with a 31% HIV seroprevalence (higher than the estimated population HIV seroprevalence of 22%) found that pathogens could be detected in 99% of stool samples collected (diarrhoeal and non-diarrhoeal patients combined), with *Cryptosporidium* spp., *Cystoisospora* spp. and *Citrobacter* spp. being significantly more common in PLHIV than HIV-uninfected individuals [5]. They found that adult PLHIV in the pre-ART era were 2.4 times as likely to suffer from diarrhoea than HIV-uninfected adults and that this increased risk lasted throughout the infection rather than being limited to those with low CD4+ cell counts [5]. Meta-analysis data indicates a high burden of *Cryptosporidium*, microsporidia and *Cystoisospora* in SSA specifically [16]. Another recent meta-analysis identified South Africa as having the highest global *Cryptosporidium* prevalence (57.0%, CI 95%: 24.4–84.5%), although they recognised that these estimates were based on limited data [27]. A study in rural South Africa found that 60% of patients suffering from chronic diarrhoea were PLHIV and that the majority of these infections were due to *Campylobacter* spp. (20%), *Plesiomonas shigelloides* (17%), *Aeromonas* spp. (13%), *Shigella* spp. (10%), *Salmonella* spp. (10%) and *Escherichia coli* spp. (10%) [28]. Testing did not include parasites or viruses and results were not stratified by ART status. Other studies in African children have identified HIV-infection as a significant risk factor for rotavirus infection [13, 29]. Very few studies in SSA in adults included screening for viruses, despite data from HICs showing that viruses (specifically norovirus and rotavirus) were responsible for as much as 44% of diarrhoea in adults presenting to emergency departments [9]. Patients with advanced HIV are at increased risk for diarrhoea due to CMV infection [30]. As CMV is mainly diagnosed on endoscopic biopsy [31], prevalence is likely underestimated.

South Africa currently has the largest PLHIV population in the world, with a prevalence of 19.5% amongst adults 15–49 years old [32]. It is estimated that 86% of PLHIV adults in South Africa have been diagnosed, with 57% of these on ART (50.8% - 72.7% by province) [33]. Women, elderly patients, those with inadequate access to WaSH facilities and those not yet initiated on ART, have the highest risk for developing diarrhoea amongst PLHIV South Africans [34]. The majority of data on HIV-related diarrhoea comes from HICs with low HIV prevalence [14]. Since diarrhoeal aetiology is likely to vary between settings [35], there is a need for more targeted epidemiological studies [18], specifically in LMICs with high HIV prevalence. There are few studies in PLHIV which investigate a comprehensive range of pathogens or infections with multiple pathogens [19]. These data are important for informing public health

interventions and updating diagnostic and treatment guidelines, especially since treatment is often based on guidelines rather than individual patient diagnostics [36]. This study aimed to determine the pathogenic causes for diarrhoeal admissions in PLHIV compared to controls without diarrhoea at three South African hospitals between July 2018 and November 2021.

## Methods

### Case and control enrolment

Patients of all ages hospitalised with diarrhoea were enrolled at Kalafong, Mapulaneng and Matikwana hospitals through the African Network for improved Diagnostics, Epidemiology and Management of common Infectious Agents (ANDEMIA) [37]. Kalafong Provincial Tertiary Hospital is based on the western outskirts of Pretoria (Gauteng Province) and serves the generally low-income communities residing in urban townships to the west of Pretoria [38]. The rural sites included Mapulaneng, a 180 bed district hospital, and Matikwana, a 250 bed regional hospital [39], both located in Bushbuckridge district in Mpumalanga Province. Only patients 5 years and older were included in the HIV analysis as there were very few PLHIV among the younger cases and controls.

Cases were defined as patients admitted with diarrhoea (three or more loose or liquid stools over a 24 hour period) for any duration. Unmatched controls were defined as individuals presenting to the hospital or clinic for reasons other than diarrhoea (vaccination clinic, orthopaedic or surgical wards), without gastrointestinal symptoms (vomiting or diarrhoea) in the past 3 weeks. The study was explained to patients fulfilling the study definition and an information leaflet provided. Written informed consent was signed by patients ≥18 years of age and by parents/guardians for patients <18 years. An assent form was signed for patients between 7–17 years. Surveillance Officers completed investigation forms by interviewing the patient, parent/guardian and reviewing clinical notes and laboratory results where available. HIV status was determined from the clinical notes and available laboratory results. Patients were enrolled within 48 hours of admission to exclude nosocomial infections. Cases were enrolled from July 2018 to June 2021 and controls enrolled from October 2019 to November 2021. In order to detect an 11% difference in pathogen detection between cases and controls (as per GEMS results [40]) at 80% power, a sample size of 450 cases and controls combined was required (assuming a 1:1 ratio of cases and controls) as calculated using chi-squared test to compare two independent proportions. Cases and controls unable to provide specimens and patients <5 years old were excluded from this analysis. Stool specimens or rectal swabs were collected from both cases and controls and transported on ice to the Centre for Enteric Diseases at the National Institute for Communicable Diseases in Johannesburg for testing.

### Laboratory testing

Pathogen screening was performed on Fast-track Diagnostics (viral and bacterial gastroenteritis and stool parasite kits) (manufactured by Siemens), Allplex (GI-Parasite and GI-Helminth (I) assays manufactured by Seegene) as well as TaqMan Array Cards (TAC, manufactured by Life Technologies) (included pathogens indicated in S1 Table). Tests were done according to the manufacturer's instructions. Monoplex PCR was used to determine final outcome where there were discrepant results between testing platforms for a specific target. All tests included internal controls for sample validation. Due to limited availability of TAC cards, cases with unknown HIV status were excluded from testing. Controls with unknown HIV status were included in the overall case-control analysis (since there were limited control specimens available) but were not included in the HIV specific case-control analysis.

## Data management and statistical analysis

Results for CD4+ cell count were extracted from the National Health Laboratory Service (NHLS) database for cases among PLHIV and were categorised into three classes (<200 cells/μl; 200–500 cells/μl; >500 cells/μl). Only CD4+ cell count results within 12 months of the enrolment were included and where multiple results were available for a single patient, the results closest to the time of enrolment were used. CD4+ cell classes were used as HIV viral load was available for very few of the PLHIV. Demographic and socioeconomic characteristics for cases and controls were compared. Clinical presentation for PLHIV versus HIV-uninfected cases were compared. Pathogen detection amongst case and control specimens were compared, stratified by HIV status. Clinical presentation and pathogen detection for cases in PLHIV was compared between CD4+ cell count categories and for ART and cotrimoxazole prophylaxis treatment. $X^2$-test and Fisher's exact test were used for categorical variables and t-test for continuous variables. *P-values*<0.05 were considered to be statistically significant. Stata software (version 14) was used for all analyses. Data are available in the supplementary materials.

## Ethical considerations

The ANDEMIA study was approved by the Human Research Ethics Committee (Medical) of the University of the Witwatersrand (approval number M170403) and the University of Pretoria Faculty of Health Sciences Research Ethics Committee (approval number 101/2017). The HIV sub-analysis was approved by the Human Research Ethics Committee (Medical) of the University of the Witwatersrand (approval numbers M190663).

# Results

## Enrolment and patient characteristics

A total of 378 specimens for patients ≥5 years were collected during the study period, including 277 cases and 101 controls (Fig 1). Amongst cases, 164 (59.2%) were PLHIV, 79 (28.5%) were HIV-uninfected and 34 (12.3%) had unknown HIV status. Amongst controls, 23 (22.8%) were PLHIV, 50 (49.5%) were HIV-uninfected and 28 (27.7%) had unknown HIV status. Cases with unknown HIV status (n = 34) were excluded (due to limited testing capacity), hence a total of 344 specimens were included in the analysis.

The median age of included patients was 34 years (IQR of 23–47), with cases being slightly older (median 36; IQR 26–51) than controls (median 31; IQR 17–38) (Table 1). The majority of patients were female (59.4%) and from the rural sites (67.4%), with proportionally more controls coming from the rural site than cases (82.2% versus 61.3%, *p*<0.001). Cases and controls were similar with respect to household crowding and dietary habits. Both cases and controls had good access to electric or gas cookers (86.1%), refrigerators (88.4%) and improved WaSH (94.8% access to improved water source and 96.4% access to private latrines/flush toilets). CD4+ cell counts were available for 99 (60.4%) of the 164 cases in PLHIV. Eight (8.1%) had counts >500 cells/μl, 21 (21.2%) were between 200–500 cells/μl and the majority (n = 70, 70.7%) had <200 cells/μl. Information regarding ART was available for 156 (95.1%) of the cases in PLHIV, of which 130 (83.3%) were on treatment. Mean CD4+ cells counts were similar between those on ART (229.23 cells/μl) and those not on ART (169.7 cells/μl), (*p* = 0.418). Information regarding cotrimoxazole treatment was available for 151 PLHIV, of which 63 (41.7%) were on treatment.

## Clinical presentation of cases

Cases among PLHIV were likely to experience symptoms for a longer duration before admission (median of 5 days versus 2 days, *p*<0.001) and were more likely to have chronic or

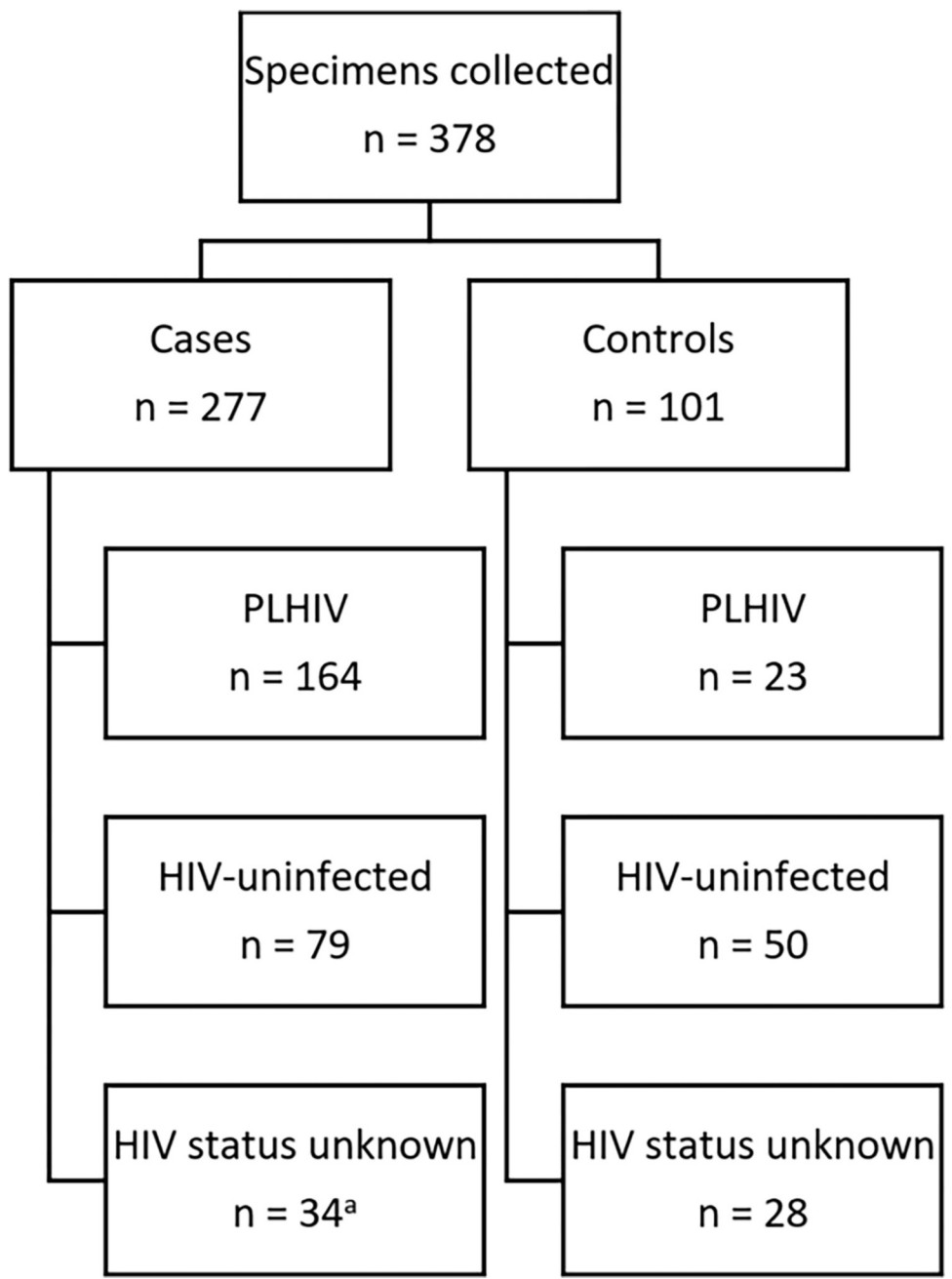

**Fig 1. Specimens included in the analysis.** [a] Due to limited TAC tests available, cases with unknown HIV status were excluded from the analysis. Controls with unknown HIV status were included to increase sample size for the overall case-control analysis, but were not included in the HIV specific analysis.

persistent diarrhoea (8.5% versus 0.0%, $p = 0.006$) than HIV-uninfected cases (Table 2). The most common symptoms experienced by all cases were fatigue (182, 74.9%), weight loss (176, 72.4%), vomiting (172, 70.8%), fever (168, 69.1%), abdominal pain (166, 68.3%) and nausea (165, 67.9%). Cases among PLHIV were more likely to suffer from weight loss (81.1% versus 54.4%, p<0.001) and nausea (72.0% versus 59.5%, $p = 0.051$), while HIV-uninfected cases were more likely to suffer from dysentery (15.2% versus 4.3%, $p = 0.003$).

**Table 1. Comparison of patient characteristics for cases and controls.**

|  | Total (n = 344) | Cases (n = 243) | Controls (n = 101) | *p*-value |
|---|---|---|---|---|
| **Age (years)–median (IQR)** | 34 (23–47) | 36 (26–51) | 31 (17–38) | **<0.001** [f] |
| **Male** | 129 (40.6%) | 88 (36.2%) | 41 (40.6%) | 0.445 |
| **Rural** | 232 (67.4%) | 149 (61.3%) | 83 (82.2%) | **<0.001** |
| **PLHIV** [a] | 187/316 (59.2%) | 164/243 (67.5%) | 23/73 (31.5%) | **<0.001** |
| **Education** [b] |  |  |  |  |
| **None** | 13/337 (3.9%) | 10/237 (4.2%) | 3/100 (3.0%) | **0.002** [g] |
| **< = 6 years** | 53/337 (15.7%) | 47/237 (19.8%) | 6/100 (6.0%) |  |
| **7–10 years** | 107/337 (31.8%) | 78/237 (32.9%) | 29/100 (29.0%) |  |
| **>10 years** | 164/337 (48.7%) | 102/237 (43.0%) | 62/100 (62.0%) |  |
| **Household crowding/people per room–mean (std. dev)** | 2.0 (1.2) | 2.0 (1.9) | 2.0 (0.9) | 0.733 [f] |
| **Electric or gas cooker** | 296 (86.1%) | 210 (86.4%) | 86 (85.1%) | 0.757 |
| **Refrigerator** | 289/327 (88.4%) | 200/231 (86.6%) | 89/96 (92.7%) | 0.115 |
| **Improved water source** [c] | 326 (94.8%) | 229 (94.2%) | 97 (96.0%) | 0.603 [g] |
| **Treat drinking water** [d] | 45 (13.1%) | 31 (12.8%) | 14 (13.9%) | 0.782 |
| **Toilet type** |  |  |  |  |
| **Private flush** | 86/337 (25.5%) | 62/241 (25.7%) | 24/96 (25.0%) | 0.604 [g] |
| **Private latrine** | 239/337 (70.9%) | 170/241 (70.5%) | 69/96 (71.9%) |  |
| **Communal flush** | 8/337 (2.3%) | 7/241 (2.9%) | 1/96 (1.0%) |  |
| **Communal latrine** | 2/337 (0.6%) | 1/241 (0.4%) | 1/96 (1.0%) |  |
| **Other** | 2/337 (0.6%) | 1/241 (0.4%) | 1/96 (1.0%) |  |
| **Dairy** [e] | 156 (45.4%) | 107 (44.0%) | 49 (48.5%) | 0.447 |
| **Deli meat** [e] | 103 (29.9%) | 79 (32.5%) | 24 (23.8%) | 0.107 |
| **Eggs** [e] | 183 (53.2%) | 127 (52.3%) | 56 (55.5%) | 0.590 |

[a] For participants with known HIV status only

[b] highest level of education that the patient (or parent/caregiver if patient is <15 years) has obtained

[c] piped water to an inside or outside tap (as opposed to river, rain or well water or water from a truck)

[d] by boiling or chemical treatment

[e] In the preceding 4 weeks

[f] t-test

[g] fisher's exact test; denominators given when different to column headings due to missing data.

### Pathogen detection in specimens of cases and controls

Pathogens were detected in 66.3% (228) of specimens tested, with significantly higher detection in cases compared with controls (72.8% versus 50.5%, *p*<0.001; Table 3). Bacteria were more prevalent in cases compared with controls (50.2% versus 28.7%, *p*<0.001), specifically *Shigella* spp., *Salmonella* spp. and *Helicobacter pylori* (24.3% versus 10.9%, *p* = 0.005; 8.6% versus 1.0%, *p* = 0.007 and 8.6% versus 0.0%, *p* = 0.001, respectively). Detection of viruses was significantly higher in cases compared with controls (21.8% versus 10.9%, *p* = 0.018), specifically for adenovirus (11.0% versus 4.0%, *p* = 0.038). Parasites were detected in 37.8% (130) of specimens tested, however, detection was not significantly higher in cases than controls (40.7% versus 30.7%, *p* = 0.080) with the available sample size. Prevalence of *Cystoisospora*, *Cryptosporidium* spp. and *Enterocytozoon* spp. was significantly higher in cases than controls (11.9% versus 0.0%, *p*<0.00; 11.1% versus 3.0%, *p* = 0.012; 4.5% versus 0.0%, *p* = 0.038, respectively). *Blastocystis* and *Schistosoma* were more prevalent amongst controls than amongst cases (23.8% versus 10.3%, *p* = 0.001; 6.9% versus 2.1%, *p* = 0.025, respectively). Cases were more likely than controls to have multiple pathogens detected (37.9% versus 22.8%, *p* = 0.007). Amongst PLHIV, *Cystoisospora* spp. prevalence was significantly

**Table 2. Clinical presentation of diarrhoeal cases.**

| | Total (n = 243) | PLHIV (n = 164) | HIV-uninfected (n = 79) | *p*-value |
|---|---|---|---|---|
| **Duration of symptoms before admission (days)–median (IQR)** | 4 (2–8) | 5 (3–8) | 2 (1–4) | <**0.001** [d] |
| **Chronic/persistent diarrhoea** [a] | 13 (5.4%) | 13 (7.9%) | 0 (0.0%) | **0.011** [e] |
| **Fatigue** | 182 (74.9%) | 124 (75.6%) | 58 (73.4%) | 0.712 |
| **Weight loss** | 176 (72.4%) | 133 (81.1%) | 43 (54.4%) | <**0.001** |
| **Vomiting** | 172 (70.8%) | 118 (72.0%) | 54 (68.4%) | 0.564 |
| **Fever** [b] | 168 (69.1%) | 113 (68.9%) | 55 (69.6%) | 0.910 |
| **Abdominal pain** | 166 (68.3%) | 106 (64.6%) | 60 (76.0%) | 0.076 |
| **Nausea** | 165 (67.9%) | 118 (72.0%) | 47 (59.5%) | 0.051 |
| **Respiratory symptoms** | 105 (43.2%) | 77 (47.0%) | 28 (35.4%) | 0.090 |
| **Headache** | 100 (41.2%) | 68 (41.5%) | 32 (40.5%) | 0.887 |
| **Chills** | 63 (25.9%) | 48 (29.3%) | 15 (19.0%) | 0.087 |
| **Arthralgia** | 51 (21.0%) | 38 (23.2%) | 13 (16.5%) | 0.229 |
| **Dermatological symptoms** | 31 (12.8%) | 25 (15.2%) | 6 (7.6%) | 0.094 |
| **Myalgia** | 24 (10.0%) | 16 (9.8%) | 8 (10.1%) | 0.928 |
| **Dysentery** [c] | 19 (7.8%) | 7 (4.3%) | 12 (15.2%) | **0.003** |
| **Neurological symptoms** | 10 (4.1%) | 6 (3.7%) | 4 (5.1%) | 0.732 [e] |
| **Painful lymphadenopathy** | 4 (1.7%) | 3 (1.9%) | 1 (1.3%) | >0.99 [e] |

[a] diarrhoea for 28 days or longer

[b] current temperature of >38˚C or history of fever in the past 10 days

[c] Dysentery defined as self-reported blood in the stool

[d] t-test

[e] fisher's exact test

higher in cases than in controls (17.7% versus 0.0%, *p* = 0.028), while *Schistosoma* was detected more often in controls compared with cases (17.4% versus 2.4%, *p* = 0.009). Amongst HIV-uninfected participants, *Shigella* spp., *Salmonella* spp. and *Helicobacter pylori* were more prevalent amongst cases than controls (36.7% versus 12.0%, *p* = 0.002; 11.4% versus 0.0%, *p* = 0.012; 10.1% versus 0.0%, *p = 0.023*). Norovirus GII, *C. difficile*, *Campylobacter*, *Cystoisospora*, *Cryptosporidium* spp. and *Enterocytozoon* spp. were more prevalent amongst PLHIV than HIV-uninfected participants (6.4% versus 1.6%, *p* = 0.032; 7.0% versus 0.8%, *p* = 0.010; 6.4% versus 0.8%, *p* = 0.018; 15.5% versus 0.01%, *p*<0.001; 15.0% versus 1.6%, *p*<0.001; 5.4% versus 0.8%, *p* = 0.031). *Shigella* spp. and *Blastocystis* were more prevalent amongst HIV-uninfected than PLHIV (27.1% versus 17.1%, *p* = 0.032; 25.6% versus 8.0%, *p*<0.001). *Shigella* spp. (18.3%) and *Cystoisospora* (17.7%) were the most prevalent pathogens detected in PLHIV diarrhoeal cases, while *Shigella* spp. (36.7%) and *Salmonella* spp. (11.4%) were the most prevalent pathogens in HIV-uninfected diarrhoeal cases. Although *Blastocystis* was prevalent in HIV-uninfected cases, it was not associated with diarrhoea as prevalence was significantly higher in HIV-uninfected controls (40.0% versus 16.5%, *p*<0.001).

## Association of CD4+ cell count and treatment with clinical presentation and pathogen detection in diarrhoeal cases among PLHIV

For cases among PLHIV, presentation did not differ by CD4+ cell count for duration of symptoms, fever, vomiting, fatigue, nausea, respiratory symptoms, abdominal pain, dermatological symptoms, chills, headache, dysentery, arthralgia, myalgia, neurological symptoms or painful lymphadenopathy (S2 Table). There was an increased proportion of cases reporting weight

**Table 3. Pathogen detection in specimens of cases and controls, stratified by HIV-infection status.**

| | Overall–n[a] (%) | | | | PLHIV–n[a] (%) | | | | HIV-uninfected–n[a] (%) | | | | p-value[c] |
|---|---|---|---|---|---|---|---|---|---|---|---|---|---|
| | Total (n = 344) | Cases (n = 243) | Controls (n = 101) | p-value[b] | Total (n = 187) | Cases (n = 164) | Controls (n = 23) | p-value[b] | Total (n = 129) | Cases (n = 79) | Controls (n = 50) | p-value[b] | |
| Any pathogen | 228 (66.3) | 177 (72.8) | 51 (50.5) | **<0.001** | 138 (73.8) | 124 (75.6) | 14 (60.9) | 0.132 | 84 (65.1) | 53 (67.1) | 31 (62.0) | 0.555 | 0.097 |
| Virus | 64 (18.6) | 53 (21.8) | 11 (10.9) | **0.018** | 43 (22.1) | 40 (24.4) | 3 (13.0) | 0.296 [e] | 20 (15.5) | 13 (16.5) | 7 (14.0) | 0.707 | 0.101 |
| Adenovirus | 31 (9.0) | 27 (11.1) | 4 (4.0) | **0.038** [e] | 20 (10.7) | 19 (11.6) | 1 (4.4) | 0.476 [e] | 10 (7.8) | 8 (10.1) | 2 (4.0) | 0.314 [e] | 0.380 |
| Ad[e]novirus 40/41 | 3 (0.9) | 3 (1.2) | 0 (0.0) | 0.558 [e] | 2 (1.1) | 2 (1.2) | 0 (0.0) | >0.99 [e] | 1 (0.8) | 1 (1.3) | 0 (0.0) | >0.99 [e] | 0.637 [e] |
| Norovirus | 18 (5.2) | 14 (5.8) | 4 (4.0) | 0.603 [e] | 13 (7.0) | 13 (7.9) | 0 (0.0) | 0.374 [e] | 5 (3.9) | 1 (1.3) | 4 (8.0) | 0.074 [e] | 0.246 |
| Norovirus GI | 4 (1.2) | 2 (0.8) | 2 (2.0) | 0.584 [e] | 1 (0.5) | 1 (0.6) | 0 (0.0) | >0.99 [e] | 3 (2.3) | 1 (1.3) | 2 (4.0) | 0.559 [e] | 0.308 [e] |
| Norovirus GII | 14 (4.1) | 12 (4.9) | 2 (2.0) | 0.248 [e] | 12 (6.4) | 12 (7.3) | 0 (0.0) | 0.367 [e] | 2 (1.6) | 0 (0.0) | 2 (4.0) | 0.148 [e] | 0.032 [e] |
| Enterovirus | 16 (4.7) | 12 (4.9) | 4 (4.0) | 0.787 [e] | 12 (6.4) | 10 (6.1) | 2 (8.7) | 0.645 [e] | 4 (3.1) | 2 (2.5) | 2 (4.0) | 0.641 [e] | 0.296 [e] |
| CMV | 3 (0.9) | 3 (1.2) | 0 (0.0) | 0.558 [e] | 3 (1.6) | 3 (1.8) | 0 (0.0) | >0.99 [e] | 0 (0.0) | 0 (0.0) | 0 (0.0) | - | 0.273 [e] |
| Astrovirus | 5 (1.5) | 4 (1.7) | 1 (1.0) | >0.99 [e] | 3 (1.6) | 3 (1.8) | 0 (0.0) | >0.99 [e] | 2 (1.6) | 1 (1.3) | 1 (2.0) | >0.99 [e] | >0.99 [e] |
| Rotavirus | 2 (0.6) | 2 (0.8) | 0 (0.0) | >0.99 [e] | 1 (0.5) | 1 (0.6) | 0 (0.0) | >0.99 [e] | 1 (0.8) | 1 (1.3) | 0 (0.0) | 1.000 [e] | >0.99 [e] |
| Sapovirus | 1 (0.3) | 1 (0.4) | 0 (0.0) | >0.99 [e] | 1 (0.5) | 1 (0.6) | 0 (0.0) | >0.99 [e] | 0 (0.0) | 0 (0.0) | 0 (0.0) | | >0.99 [e] |
| >1 virus detected | 9 (2.6) | 7 (2.9) | 2 (2.0) | >0.99 [e] | 7 (3.7) | 7 (4.3) | 0 (0.0) | 0.600 [e] | 2 (1.6) | 0 (0.0) | 2 (4.0) | 0.148 [e] | 0.318 [e] |
| Bacteria | 151 (48.9) | 122 (50.2) | 29 (28.7) | **<0.001** | 87 (46.5) | 76 (46.3) | 11 (47.8) | 0.894 | 61 (47.3) | 46 (58.2) | 15 (30.0) | **0.002** | 0.894 |
| *Shigella* spp. | 70 (20.4) | 59 (24.3) | 11 (10.9) | **0.005** | 32 (17.1) | 30 (18.3) | 2 (8.7) | 0.378 [e] | 35 (27.1) | 29 (36.7) | 6 (12.0) | **0.002** | **0.032** |
| *Salmonella* spp. | 22 (6.4) | 21 (8.6) | 1 (1.0) | **0.007** [e] | 13 (7.0) | 12 (7.3) | 1 (4.4) | >0.99 [e] | 9 (7.0) | 9 (11.4) | 0 (0.0) | **0.012** [e] | 0.993 |
| *C. difficile* | 14 (4.1) | 13 (5.4) | 1 (1.0) | 0.074 [e] | 13 (7.0) | 12 (7.3) | 1 (4.4) | >0.99 [e] | 1 (0.8) | 1 (1.3) | 0 (0.0) | >0.99 [e] | **0.010** [e] |
| *Campylobacter* | 13 (3.8) | 12 (4.9) | 1 (1.0) | 0.119 [e] | 12 (6.4) | 12 (7.3) | 0 (0.0) | 0.367 [e] | 1 (0.8) | 0 (0.0) | 1 (2.0) | 0.388 [e] | **0.018** [e] |
| Shiga toxin-producing *E. coli* (STEC) | 2 (0.6) | 0 (0.0) | 2 (2.0) | 0.086 [e] | 0 (0.0) | 0 (0.0) | 0 (0.0) | | 2 (1.6) | 0 (0.0) | 2 (4.0) | 0.148 [e] | 0.166 [e] |
| Enterotoxigenic *E. coli* (ETEC) | 11 (3.2) | 9 (3.7) | 2 (2.0) | 0.519 [e] | 7 (3.7) | 5 (3.1) | 2 (8.7) | 0.207 [e] | 4 (3.1) | 4 (5.1) | 0 (0.0) | 0.157 [e] | >0.99 [e] |
| Enteropathogenic *E. coli* (EPEC) | 21 (6.1) | 14 (5.8) | 7 (6.9) | 0.631 | 15 (8.0) | 11 (6.7) | 4 (17.4) | 0.094 [e] | 6 (4.7) | 3 (3.8) | 3 (6.0) | 0.676 [e] | 0.237 |
| Enteroaggregative *E. coli* (EAEC) | 41 (11.9) | 28 (11.5) | 13 (12.9) | 0.725 | 25 (13.4) | 20 (12.2) | 5 (21.7) | 0.208 | 14 (10.9) | 8 (10.1) | 6 (12.0) | 0.739 | 0.504 |
| *E. coli* O157 | 4 (1.2) | 4 (1.7) | 0 (0.0) | 0.325 [e] | 3 (1.6) | 3 (1.8) | 0 (0.0) | >0.99 [e] | 1 (0.8) | 1 (1.3) | 0 (0.0) | >0.99 [e] | 0.648 [e] |
| *Plesiomonas* | 1 (0.3) | 1 (0.4) | 0 (0.0) | >0.99 [e] | 1 (0.5) | 1 (0.6) | 0 (0.0) | >0.99 [e] | 0 (0.0) | 0 (0.0) | 0 (0.0) | | >0.99 [e] |
| *Helicobacter pylori* | 21 (6.1) | 21 (8.6) | 0 (0.0) | **0.001** [e] | 13 (7.0) | 13 (7.9) | 0 (0.0) | 0.374 [e] | 8 (6.2) | 8 (10.1) | 0 (0.0) | **0.023** [e] | >0.99 |
| >1 bacteria detected | 45 (13.1) | 37 (15.2) | 8 (7.9) | 0.067 | 23 (12.3) | 20 (12.2) | 3 (13.0) | >0.99 [e] | 11 (8.5) | 8 (10.1) | 3 (6.0) | 0.528 [e] | 0.287 |
| Parasite[d] | 130 (37.8) | 99 (40.7) | 31 (30.7) | 0.080 | 85 (45.5) | 78 (47.6) | 7 (30.4) | 0.122 | 43 (33.3) | 21 (26.6) | 22 (44.0) | 0.042 | **0.031** |
| *Cystoisospora* | 29 (8.4) | 29 (11.9) | 0 (0.0) | **<0.001** [e] | 29 (15.5) | 20 (17.7) | 0 (0.0) | **0.028** [e] | 0 (0.0) | 0 (0.0) | 0 (0.0) | | **<0.001** [e] |

(*Continued*)

**Table 3.** (Continued)

| | Overall–nᵃ (%) | | | | PLHIV–nᵃ (%) | | | | HIV-uninfected–nᵃ (%) | | | | p-value C |
|---|---|---|---|---|---|---|---|---|---|---|---|---|---|
| | Total (n = 344) | Cases (n = 243) | Controls (n = 101) | p-value b | Total (n = 187) | Cases (n = 164) | Controls (n = 23) | p-value b | Total (n = 129) | Cases (n = 79) | Controls (n = 50) | p-value b | |
| *Cryptosporidium* spp. | 30 (8.7) | 27 (11.1) | 3 (3.0) | **0.012** e | 28 (15.0) | 26 (15.9) | 2 (8.7) | 0.537 e | 2 (1.6) | 1 (1.3) | 1 (2.0) | >0.99 e | **<0.001** e |
| *Blastocystis* | 49 (14.2) | 25 (10.3) | 24 (23.8) | **0.001** | 15 (8.0) | 12 (7.3) | 3 (13.0) | 0.403 e | 33 (25.6) | 13 (16.5) | 20 (40.0) | **0.003** | **<0.001** |
| *Giardia* spp. | 12 (3.5) | 9 (3.7) | 3 (3.0) | >0.99 e | 7 (3.7) | 7 (4.3) | 0 (0.0) | 0.600 e | 4 (3.1) | 2 (2.5) | 2 (4.0) | 0.641 e | >0.99 e |
| *Enterocytozoon* spp. | 11 (3.2) | 11 (4.5) | 0 (0.0) | **0.038** e | 10 (5.4) | 10 (6.1) | 0 (0.0) | 0.614 e | 1 (0.8) | 1 (1.3) | 0 (0.0) | >0.99 e | **0.031** e |
| *Schistosoma* | 12 (3.5) | 5 (2.1) | 7 (6.9) | **0.025** | 8 (4.3) | 4 (2.4) | 4 (17.4) | **0.009** e | 4 (3.1) | 1 (1.3) | 3 (6.0) | 0.298 e | 0.767 e |
| *Dientamoeba* | 2 (0.6) | 1 (0.4) | 1 (1.0) | 0.502 e | 0 (0.0) | 0 (0.0) | 0 (0.0) | | 2 (1.6) | 1 (1.3) | 1 (2.0) | >0.99 e | 0.166 e |
| >1 parasite detected | 19 (5.5) | 13 (5.4) | 6 (5.9) | 0.827 | 24 (12.8) | 22 (13.4) | 2 (8.7) | 0.744 e | 10 (7.8) | 6 (7.6) | 4 (8.0) | >0.99 e | 0.152 |
| **Mixed infections** Virus-bacteria | 40 (11.6) | 34 (14.0) | 6 (5.9) | **0.034** | 25 (13.4) | 23 (14.0) | 2 (8.7) | 0.744 e | 14 (10.9) | 10 (12.7) | 4 (8.0) | 0.564 e | 0.504 |
| Virus-parasite | 35 (10.2) | 29 (11.9) | 6 (5.9) | 0.094 | 25 (13.4) | 23 (14.0) | 2 (8.7) | 0.744 e | 11 (8.5) | 7 (8.9) | 4 (8.0) | >0.99 e | 0.183 |
| Bacteria-parasite | 72 (20.9) | 60 (24.7) | 12 (11.9) | **0.008** | 46 (24.6) | 41 (25.0) | 5 (21.7) | 0.734 | 23 (17.8) | 16 (20.3) | 7 (14.0) | 0.366 | 0.152 |
| >1 pathogen detected | 115 (33.4) | 92 (37.9) | 23 (22.8) | **0.007** | 73 (39.0) | 66 (40.2) | 7 (30.4) | 0.366 | 40 (31.0) | 26 (32.9) | 14 (28.0) | 0.557 | 0.143 |

ᵃ Total is 344 for overall analysis and 316 for the HIV analysis due to 28 controls with unknown HIV-status being excluded from the HIV analysis

ᵇ p-value comparing pathogen detection in cases versus controls (for PLHIV and HIV-uninfected groups respectively)

C p-value comparing pathogen detection in PLHIV versus HIV-uninfected persons (cases and controls combined)

ᵈ The following parasites were screened for but not detected: *Vibrio cholerae*, *Yersinia enterocolitica*, *E. histolytica*, *Strongyloides* spp., *Cyclospora*, *Hymenolepsis*, *Ascaris*, *Taenia*, *Trichuris*, *Ancylostoma*, *Enterobius*, *Necator*

ᵉ Fisher's exact test

loss with lower CD4+ cell counts (91.4%, 81.0% and 62.5% for CD4+ cell counts of <200, 200–500, >500 cells/mm³ respectively; $p = 0.032$). Pathogens were detected more frequently in cases with low CD4+ cell counts (75.7%, 42.9% and 37.5% for CD4+ cell counts of <200, 200–500, >500 cells/mm³ respectively; $p = 0.004$) (S3 Table). *Cryptosporidium* spp. specifically, was detected more often in those with low CD4+ cell counts (25.7%, 4.8% and 0.0% for CD4+ cell counts of <200, 200–500, >500 cells/mm³ respectively; $p = 0.039$).

Clinical presentation was similar amongst cases in PLHIV on ART and those not on ART (S4 Table). Those on cotrimoxazole prophylaxis were more likely to experience fatigue (88.9% versus 69.3%, $p = 0.005$), weight loss (90.5% versus 76.1%, $p = 0.023$) and chills (50.8% versus 15.9%, $p<0.001$) then those not on cotrimoxazole. Pathogen detection was similar amongst cases in PLHIV on ART and those not on ART (72.3% versus 88.5%, $p = 0.082$) (S5 Table). Parasites, specifically *Cystoisospora* spp. were more commonly detected amongst patients on cotrimoxazole compared with those not on cotrimoxazole (27.0% versus 11.4%, $p = 0.014$). Other pathogens were found in similar proportions across treatment groups.

## Clostridioides difficile detection

*C. difficile* was detected in 14 cases, 13 of which were in PLHIV (92.9%). Symptoms were similar between cases in which *C. difficile* was detected and those without *C. difficile* detection,

however those with *C. difficile* detected were more likely to have had a longer duration of illness before admission (6.5 (IQR 4–11) days versus 4 (IQR 2–7) days). It is important to distinguish *C. difficile* colonisation from disease to avoid unnecessary antimicrobial therapy. The case definition, as per the guidelines for diagnostic and clinical management of *C. difficile* from the South African Society for Clinical Microbiology, includes onset of diarrhoea more than 48 hours after admission, or diarrhoea that continues for 3 days post admission and where there is no likely alternatively cause (no other pathogens detected or use of laxatives), or where the patient has had an admission or antibiotics use in the past 12 weeks [41]. Three of the 14 cases did not match this case definition as they did not have an admission in the past 12 weeks and had multiple pathogens detected on PCR. These three cases were however in PLHIV. Five of the cases had no other pathogens detected on PCR, all of which were in PLHIV. Eight of the cases did have a prior admission, however the majority of these (n = 6, 75.0%) had multiple pathogens detected on PCR. In addition, the guidelines recommend that specimens testing positive for *C. difficile* on PCR (which is highly sensitive for detection of C. *difficile*), should have further toxins A/B immunoassays performed (which have high specificity for diagnosis of *C. difficile* infection). This was not done in the current analysis.

## Discussion

Despite advances in HIV treatment, diarrhoea amongst PLHIV remains a significant public health challenge [19]. Our study found that the majority (67.5%) of patients 5 years and older with known HIV-status admitted to the study sites with diarrhoea were in PLHIV, highlighting the importance of HIV-related illnesses among South African adults hospitalised with diarrhoea. The Joint United Nations Programme on HIV/AIDS (UNAIDS) 90-90-90 treatment target aims to ensure that 90% of PLHIV are diagnosed, 90% of those diagnosed are initiated on ART and 90% of those on ART should achieve viral suppression [42]. A recent South African analysis estimated that 70.7% of those with diagnosed HIV-infection have initiated ART, and 87.4% of these are thought to be virally suppressed [42]. Proportions reported in the current analysis reflect these numbers, as 83.3% of patients with diagnosed HIV infection and known ART status were on treatment, however a large proportion of PLHIV patients included in this analysis had CD4+ cell count <200 cells/µl (70/99, 70.7%) indicating that they are unlikely to be virally suppressed. Mpumalanga (home province to the majority of the patients included in this analysis) is the only South African province in which the third-90 indicator value (the percentage of those on ART that are virally suppressed) is below 85% [42]. It is also likely that the study design was skewed towards detection of poorly suppressed HIV cases as they are most likely to be hospitalised for diarrhoea. Diarrhoea in patients with viral suppression will likely be seen at a clinic level. Further studies at clinic level are recommended in order to include a broader spectrum of cases. The clinical presentation of diarrhoea amongst cases in PLHIV differs to that of HIV-uninfected cases. Diarrhoeal cases among PLHIV were more likely to have symptoms for a longer duration before admission and to suffer from weight loss and nausea. HIV-uninfected cases were more likely to suffer from dysentery. This finding has not been reported in other studies, however it is likely related to aetiological differences between PLHIV and HIV-uninfected diarrhoeal patients. Amongst cases in PLHIV, patients on cotrimoxazole were more likely to experience fatigue, weight loss and chills and were more likely to have parasites detected, specifically *Cystoisospora* spp., compared to those not on cotrimoxazole. This is an interesting finding since cotrimoxazole is the prescribed treatment for this pathogen. Since all patients with CD4+ <200 cells/µl should receive cotrimoxazole prophylaxis, it can be used as a surrogate marker for advanced HIV disease. However this finding may also point to changes in *Cystoisospora* sensitivity for cotrimoxazole or may

indicate that these patients are carriers or take longer to clear the pathogen. The possibility of drug resistance should be further investigated in future studies.

The use of molecular methods and the expanded panel of pathogens included in testing resulted in a high pathogen detection rate, with pathogen detection being significantly higher in cases compared with controls. Although viruses were commonly detected (18.5% of all specimens tested) and were more likely to be detected in cases as compared with controls, when stratified by HIV status, viruses were not significantly associated with diarrhoea in this study. Although this lack of association may be partly due to asymptomatic carriage of viruses (specifically in HIV-uninfected patients), it is likely that the sample size was insufficient to determine significant differences between cases and controls when stratified by HIV status. Interestingly, norovirus GII was more prevalent in PLHIV than HIV-uninfected individuals and detection in PLHIV was limited to cases. Previous findings from South Africa have shown that children living with HIV are more likely to suffer poor outcomes related to norovirus infection [43], however detection rates were similar amongst with and without HIV-infection. This increased detection of norovirus in PLHIV ≥5 years of age has not been previously described. Bacteria was detected in 43.9% of specimens tested and was significantly associated with diarrhoea overall and amongst HIV-uninfected patients. Interestingly, prevalence of bacterial pathogens was high in both cases and controls within the PLHIV group. This was specifically due to the high prevalence of *E. coli* spp. (including ETEC, EPEC and EAEC) amongst controls with HIV infection. High *E. coli* spp. colonisation rates in PLHIV have been documented in other settings [44]. We found *Shigella* spp, *Salmonella* spp. and *H. pylori* to be significantly associated with diarrhoea in the current study, specifically amongst HIV-uninfected patients. It's likely that the sample size was insufficient to detect a difference in prevalence between cases and controls in the HIV-uninfected group. *Campylobacter* spp. and *C. difficile* were detected more often in PLHIV as compared with HIV-uninfected patients but were not significantly associated with diarrhoea (likely due to small sample sizes).

As expected from literature [14], parasite prevalence was significantly higher amongst PLHIV compared with HIV-uninfected patients, specifically *Cystoisospora*, *Cryptosporidium* spp. and *Enterocytozoon* spp., although only *Cystoisospora* was significantly associated with diarrhoea amongst PLHIV. The case-control analysis indicated that *Blastocystis* and *Schistosoma* are unlikely to be causative agents of diarrhoea since these pathogens were detected at higher rates in controls than in cases. Prevalence of *Schistosoma* among healthy individuals has been reported to be as high as 51.2% in the Democratic Republic of Congo and is likely to be higher in villages and rural areas where water is collected from rivers and dam [45]. *Blastocystis* has been noted as an important cause of diarrhoea amongst immunosuppressed patients [46], however this was not found in the current study.

Diarrhoeal aetiology differed with HIV status, with *Shigella* spp. and *Salmonella* spp. being prevalent amongst HIV-uninfected cases and *Shigella* spp., *Cystoisospora*, and *Cryptosporidium* spp. being prevalent amongst cases in PLHIV. These differences should be considered during development of diagnostic and treatment guidelines. This study highlights the importance of *Shigella* spp. in diarrhoeal morbidity amongst adults (regardless of HIV status). According to the Global Burden of Disease Study (GBD), *Shigella* spp. were the leading cause of diarrhoeal deaths amongst individuals over the age of 5 years and the second leading cause of death in young children in 2016 [1]. While there is currently no approved *Shigella* vaccine, there are several candidates which show promise for efficacy testing [47]. In order to introduce a candidate vaccine, burden estimates are required. Although a larger sample size is required, this analysis gives an indication of the high burden of *Shigella* spp. in our setting. Further work is required to determine if PLHIV should be considered as a risk group for possible target should *Shigella* vaccine become available.

Our results are in line with clinical literature which indicates a shift in diarrhoeal aetiology with CD4+ cell count amongst PLHIV. Pathogen detection was high in those with CD4+ cell counts <200 cells/μl, indicating increased opportunistic infections (specifically *Cryptosporidium* spp.). Pathogens could only be detected in just over a third of cases with high CD4+ cell count (>500 cells/μl), suggesting that these cases are possibly drug-related. This finding may be useful for treatment guidelines. It also highlights the importance of addressing diarrhoea in patients on ART especially since chronic diarrhoea has been identified as an major cause of dropout from ART care services [48].

Detection of *C. difficile* in this study was almost exclusive to PLHIV (13/14, 92.9%; including 12 cases and 1 control). Data from another hospital study in Gauteng Province estimated that community acquired *C. difficile* infections represent only 1.3% of cases (with the remaining 98.7% being hospital acquired) [49]. Our study design excluded cases of hospital-acquired diarrhoea, as enrolment and specimen collection was done within 48 hours of admission. The 5.4% *C. difficile* prevalence reported here is lower than systematic review estimates of 15.8% amongst symptomatic non-immunosuppressed in-patients from LMIC [50]. This systematic review indicated similar rates amongst immunosuppressed and non-immunosuppressed populations [50] whereas the current study found that *C. difficile* was more commonly detected in cases among PLHIV than in–uninfected cases. While not considered an opportunistic infection, *C. difficile* is associated with exposure to healthcare settings and with the use of antibiotics, both of which are increased amongst PLHIV [51]. The increased management of HIV patients in an outpatient setting [52] may dispute the inclusion of recent admission as a criterion in the case definition. There is also growing evidence to suggest the possibility of *C. difficile* being an important community-acquired pathogen [52], and consideration of this should be given in populations with high HIV prevalence.

A major limitation of this study was the small sample size which restricted the analysis (attributable fractions and odds ratios could not be calculated as many categories had zero count). The sample size particularly limited the comparison of clinical presentation and pathogen detection between CD4+ categories and treatment groups amongst PLHIV. The ANDEMIA study was not specifically aimed at enrolling cases and controls amongst PLHIV, therefore this analysis relied on incidental HIV enrolments. Enrolment, specifically of controls, was also hampered by COVID-19 restrictions. An extension of this analysis would benefit from enrolling controls from HIV clinics. Viral load was not analysed here, since few recent results were available; thus recent CD4+ cell count was used as a proxy. CD4+ cell counts were not available for all cases in PLHIV at the time of admission. Many of those without CD4+ cell counts available were matched from NHLS data, however these tests were not always performed at the same time as the enrolment. We used a cut off of 12 months on either side of enrolment for CD4+ cell counts as this was deemed to be a reasonable estimation of the CD4 + cell count at admission. In cases where there were significant changes in CD4+ cell counts over time, these may have been incorrectly quantified. There were also limited data available for antibiotic use. Date of last dose was available, however start date for antibiotic course was not collected. This has limitations specifically for the *C. difficile* analysis, since we could not determine the length of antibiotic courses. Future studies should also investigate the expansion of diagnostic testing for *M. avium* which was not done in the present study. This study was subject to limitations inherent to case control studies, including recall bias for risk factors and selection bias. The use of community controls may have minimised selection bias, however the requirements for control enrolment included minimising additional costs and disruption to case enrolment and necessitated that enrolment strategies be comparable across all ANDEMIA network sites.

In conclusion, this study highlights the importance of HIV-related diarrhoea amongst South African inpatients ≥5 years of age, and underscores the importance of research in a high HIV prevalence setting in order to improve the understanding of aetiology of diarrhoea. These data are important for development of guidelines and treatment protocols as well as for preventative interventions, such as vaccine introduction. Our data specifically highlights the importance of *Shigella* spp. amongst both PLHIV and HIV-uninfected diarrhoeal patients. We suggest that research be expanded to primary healthcare levels in order to include a broader spectrum of HIV disease, specifically to investigate diarrhoea in individuals with well-controlled HIV.

## Supporting information

**S1 Data.**
(DTA)

**S1 Table. Pathogens included in molecular testing.**
(DOCX)

**S2 Table. Clinical presentation for cases among PLHIV, stratified by CD4+ cell counts.**
(DOCX)

**S3 Table. Pathogen detection in specimens of cases among PLHIV, stratified by CD4+ cell count.**
(DOCX)

**S4 Table. Clinical presentation of cases among PLHIV, stratified by treatment.**
(DOCX)

**S5 Table. Pathogens detected in cases among PLHIV, stratified by treatment.**
(DOCX)

## Acknowledgments

We wish to acknowledge the ANDEMIA participants, funders, staff and clinical staff at the hospitals as well as patients that took part in the study. Thank you to Ann Christin Vietor for statistical assistance.

## Author Contributions

**Conceptualization:** Siobhan L. Johnstone, Linda Erasmus, Juno Thomas, Michelle J. Groome, Nicola A. Page.

**Data curation:** Siobhan L. Johnstone, Nicolette M. du Plessis, Theunis Avenant, Maryke de Villiers, Nicola A. Page.

**Formal analysis:** Siobhan L. Johnstone, Linda Erasmus, Juno Thomas, Michelle J. Groome, Nicola A. Page.

**Funding acquisition:** Nicola A. Page.

**Investigation:** Siobhan L. Johnstone.

**Methodology:** Siobhan L. Johnstone, Juno Thomas, Michelle J. Groome, Nicola A. Page.

**Project administration:** Siobhan L. Johnstone, Nicola A. Page.

**Writing – original draft:** Siobhan L. Johnstone.

**Writing – review & editing:** Linda Erasmus, Juno Thomas, Michelle J. Groome, Nicolette M. du Plessis, Theunis Avenant, Maryke de Villiers, Nicola A. Page.

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
