## [Decision Letter · Decision Letter 0]

30 Mar 2023

PGPH-D-23-00297

Epidemiology and aetiology of moderate to severe diarrhoea in hospitalised HIV-infected patients ≥5 years old in South Africa, 2018-2021: a case-control analysis

Dear Dr. Johnstone,

Thank you for submitting your manuscript to PLOS Global Public Health. After careful consideration, we feel that it has merit but does not fully meet PLOS Global Public Health’s publication criteria as it currently stands. Therefore, we invite you to submit a revised version of the manuscript that addresses the points raised during the review process.

Three independent reviewers have assessed the manuscript. All the reviews agree that this work deserves publication in PLOS Global Public Health. However, they have also raised some minor concerns that should be addressed before the manuscript can be accepted for publication. Please pay special attention to the comment by reviewer three regarding data availability.

We look forward to receiving your revised manuscript.

Kind regards,

Chrispin Chaguza, Ph.D

Academic Editor

Journal Requirements:

2. Please provide separate figure files in .tif or .eps format only and remove any figures embedded in your manuscript file. Please also ensure that all files are under our size limit of 10MB.

3. We noticed that you used "data not shown" in the manuscript. We do not allow these references, as the PLOS data access policy requires that all data be either published with the manuscript or made available in a publicly accessible database. Please amend the supplementary material to include the referenced data or remove the references.

4. We notice that your supplementary tables are included in the manuscript file. Please remove them and upload them with the file type 'Supporting Information'. Please ensure that each Supporting Information file has a legend listed in the manuscript after the references list.

5. In the online submission form, you indicated that "The datasets generated and/or analysed during the current study are not publicly available as they include study data which has not yet been published. Data are

available from the corresponding author on reasonable request". All PLOS journals now require all data underlying the findings described in their manuscript to be freely available to other researchers, either 1. In a public repository, 2. Within the manuscript itself, or 3. Uploaded as supplementary information.

Additional Editor Comments (if provided):

Please include a statement on the determination of the sample size for the study in the methods section.

Reviewers' comments:

Reviewer's Responses to Questions

**Comments to the Author**

1. Does this manuscript meet PLOS Global Public Health’s publication criteria? Is the manuscript technically sound, and do the data support the conclusions? The manuscript must describe methodologically and ethically rigorous research with conclusions that are appropriately drawn based on the data presented.

Reviewer #1: Yes

Reviewer #2: Yes

Reviewer #3: Partly

2. Has the statistical analysis been performed appropriately and rigorously?

Reviewer #1: Yes

Reviewer #2: Yes

Reviewer #3: Yes

3. Have the authors made all data underlying the findings in their manuscript fully available (please refer to the Data Availability Statement at the start of the manuscript PDF file)?

Reviewer #1: No

Reviewer #2: No

Reviewer #3: No

4. Is the manuscript presented in an intelligible fashion and written in standard English?

Reviewer #1: Yes

Reviewer #2: Yes

Reviewer #3: Yes

5. Review Comments to the Author

Reviewer #1: Thank you very much for letting me review this paper by Johnstone et al. The study provides important insights into the pathogenic causes of diarrhoeal admissions in HIV-infected patients in South Africa. The study design and methods are appropriate and well-described, and the sample size is reasonable given the scope of the study. However, there are a few minor limitations and areas for improvement that should be considered.

• The authors use ‘HIV-infected’ to refer to people living with HIV, please note that this is considered (and is) discriminatory language. Please replace this with the term "people living with HIV" (PLHIV). This is considered more appropriate and person-centered language than "HIV-infected." The latter term focuses solely on the medical condition and can perpetuate stigma and discrimination towards individuals living with HIV. Non-PLHIV can simply be referred to as HIV-Negative.

• Line 146 starts with saying patients of all ages were enrolled but later on text suggests this was only >5 years, or rather those included in this particular study, please clarify this by adding a sentence that the data obtained from the ANDEMIA study for this analysis focused on >5 years. This correction will also benefit from an explanation on why < 5 were excluded. Line 155, what about the 5 and 6 year olds?

• Considering the importance of viral load in HIV studies, this reviewer only landed on the explanation about why viral load was not available for analysis later in the discussion. Please include this information in the methods section soon after you mention CD4, considering the first few paragraphs of the discussion use viral suppression to discuss findings on CD4

• Statistically speaking, you can not use the word ‘Effects’ in lines 265-266 for a CC study as you are not trying to establish causality with this study. Table S2 presents chisq associations and they should be referred to as such. Causal language should be avoided.

• If this reviewer understood this very well, there were no differences in clinical presentation (Lines 267-269) between the different categories of CD4 and between treated and untreated, is there an explanation for this? Considering the untreated and those with low CD4 are expected to present worse?

• Still on above, while there was no difference in clinical presentations, there were differences in pathogen prevalence (i.e. more pathogens in PLHIV with low CD4,Lines 274-276), how should the two be married together? Keep in mind a layman reader.

• The discussion could also benefit from acknowledgement of the limitations of the case control approach.

• Please make sure all the abbreaviations have been defined STEC,ETEC,EPEC etc.

Reviewer #2: The authors have addressed an important area on understanding the etiology of diarrhea in HIV-infected adults. Big studies on the etiology of diarrhea in SSA has been done for the under-five age group and not adults. Although the sample size limited their analysis, the results presented are important.

I recommend the manuscript for publication with minor corrections on the basis that the methodology section has not been presented in sufficient detail for the study to be replicated. More information or citations should be included for the following:

Stool specimen/ rectal swab collection? who collected and how was the pre-analytical sample treatment done to ensure the integrity of the specimen.

Explain or cite the three laboratory tests that were done.

Reviewer #3: This is a carefully done analysis that adressess important questions about the aetiology of diarrhea in high-HIV prevalence settings. However, I am not sure that it currently complies with the journal's data availability policy (https://journals.plos.org/globalpublichealth/s/data-availability). For reference, the author's data availability statement is: "The datasets generated and/or analysed during the current study are not publicly available as they include study data which has not yet been published. Data are available from the corresponding author on reasonable request." This conflicts with the journal's policy, which states "it is not acceptable for an author to be the sole named individual responsible for ensuring data access." Furthermore, the potential for future publications is not considered a valid reason to not share data under the journal's policy (see the "Unacceptable Data Access Restrictions" subsection). The authors should review and respond to both the journal's data availability policy and my suggestions below before resubmitting.

The abstract seems to be split into different sections/paragraphs corresponding to the Background, Methods, Results, and Conclusion. If these sections are truly intended to stand apart in the abstract as separate paragraphs, they should be labeled with subheaders.

In the last paragraph of the introduction, it may be useful to state the seroprevalence of HIV infection in South African adults, to further support the statement "South Africa currently has the highest HIV burden in the world" and enable comparisons with the figures given for Botswana and Zambia.

The methods section should clarify that controls and cases were not explicitly matched on the basis of age, location, HIV-status, or any other characteristics.

Minor grammer and typographical suggestions:

On line 33, I would suggest inserting the word "was" between "screening" and "performed".

On line 58, "With" may be a more suitable word choice than "Despite" (if WaSH improvements led to a decline in disease from water-borne pathogens, we would expect food-borne pathogens to comprise a higher share of overall diarrheal illness). Furthermore, since "WaSH" is an umbrella term that covers not just clean drinking water but also interventions that should prevent food-borne illness (hand washing, segregation of sewage from agriculture, etc), maybe the term "WaSH" should be replaced with "access to safe drinking water" or something similar, if that's what the authors mean.

6. PLOS authors have the option to publish the peer review history of their article (what does this mean?). If published, this will include your full peer review and any attached files.

**Do you want your identity to be public for this peer review?** For information about this choice, including consent withdrawal, please see our Privacy Policy.

Reviewer #1: No

Reviewer #2: No

Reviewer #3: **Yes: **Jo Walker

---

## [Decision Letter · Decision Letter 1]

8 Jun 2023

PGPH-D-23-00297R1

Epidemiology and aetiology of moderate to severe diarrhoea in hospitalised patients ≥5 years old living with HIV in South Africa, 2018-2021: a case-control analysis

Dear Dr. Johnstone,

Thank you for submitting your manuscript to PLOS Global Public Health. After careful consideration, we feel that it has merit but does not fully meet PLOS Global Public Health’s publication criteria as it currently stands. Therefore, we invite you to submit a revised version of the manuscript that addresses the points raised during the review process.

The reviewers have assessed your revised manuscript again, and they generally favour its publication in PLOS Global Public Health. However, before formally accepting the manuscript, we would like you to address a few minor comments by reviewer 3.

We look forward to receiving your revised manuscript.

Kind regards,

Chrispin Chaguza, Ph.D

Academic Editor

Journal Requirements:

Additional Editor Comments (if provided):

Reviewers' comments:

Reviewer's Responses to Questions

**Comments to the Author**

1. If the authors have adequately addressed your comments raised in a previous round of review and you feel that this manuscript is now acceptable for publication, you may indicate that here to bypass the “Comments to the Author” section, enter your conflict of interest statement in the “Confidential to Editor” section, and submit your "Accept" recommendation.

Reviewer #2: All comments have been addressed

Reviewer #3: (No Response)

2. Does this manuscript meet PLOS Global Public Health’s publication criteria? Is the manuscript technically sound, and do the data support the conclusions? The manuscript must describe methodologically and ethically rigorous research with conclusions that are appropriately drawn based on the data presented.

Reviewer #2: Yes

Reviewer #3: Yes

3. Has the statistical analysis been performed appropriately and rigorously?

Reviewer #2: Yes

Reviewer #3: Yes

4. Have the authors made all data underlying the findings in their manuscript fully available (please refer to the Data Availability Statement at the start of the manuscript PDF file)?

Reviewer #2: Yes

Reviewer #3: Yes

5. Is the manuscript presented in an intelligible fashion and written in standard English?

Reviewer #2: Yes

Reviewer #3: Yes

6. Review Comments to the Author

Reviewer #2: In the methods section, indicate if "tests were done according to manufacturer’s instruction" for the test kits used.

if there were any modifications on the manufacturer's instructions then they should be spelled out., otherwise "tests were done according to manufacturer’s instruction" should be indicated.

I am however very satisfied with the revised version.

Reviewer #3: The author's revisions have definitely improved the paper. I only have a few minor suggestions, and recommend publication if these are addressed.

On line 129 in the intro, “highest HIV burden” should be changed to “largest PLHIV population” (if that is what the authors mean) to avoid confusion, as the previous paragraph cites slightly higher prevalence estimates for Botswana and Zambia (23.9% and 22% vs 19.5%).

The new sample size statement in the methods section is appreciated, but it should be amended to 1) show how this calculation was performed or cite the method/approach used 2) clarify whether the requirement is 450 cases and controls combined, or 450 of each. or 450 total. If the former, the authors should clarify whether their power calculation assumes a 1:1 ratio of cases and controls (a 225:225 split will obviously have more power than a 449:1 split).

At a relevant point in the text (possibly the “data management and statistical analysis” section of the methods), the authors should include a short sentence indicating that the data is available in the supplementary materials.

7. PLOS authors have the option to publish the peer review history of their article (what does this mean?). If published, this will include your full peer review and any attached files.

**Do you want your identity to be public for this peer review?** For information about this choice, including consent withdrawal, please see our Privacy Policy.

Reviewer #2: **Yes: **Angeziwa Chunga Chirambo

Reviewer #3: **Yes: **Joseph Walker

---

## [Editor Report · Decision Letter 2]

7 Aug 2023

Epidemiology and aetiology of moderate to severe diarrhoea in hospitalised patients ≥5 years old living with HIV in South Africa, 2018-2021: a case-control analysis

PGPH-D-23-00297R2

Dear Ms Johnstone,

We are pleased to inform you that your manuscript 'Epidemiology and aetiology of moderate to severe diarrhoea in hospitalised patients ≥5 years old living with HIV in South Africa, 2018-2021: a case-control analysis' has been provisionally accepted for publication in PLOS Global Public Health.

Best regards,

Chrispin Chaguza, Ph.D

Academic Editor
